# INTERPOLATION-PREDICTION NETWORKS FOR IRREGULARLY SAMPLED TIME SERIES

**Satya Narayan Shukla**
College of Information and Computer Sciences
University of Massachusetts Amherst
snshukla@cs.umass.edu

**Benjamin M. Marlin**
College of Information and Computer Sciences
University of Massachusetts Amherst
marlin@cs.umass.edu

## ABSTRACT

In this paper, we present a new deep learning architecture for addressing the problem of supervised learning with sparse and irregularly sampled multivariate time series. The architecture is based on the use of a semi-parametric interpolation network followed by the application of a prediction network. The interpolation network allows for information to be shared across multiple dimensions of a multivariate time series during the interpolation stage, while any standard deep learning model can be used for the prediction network. This work is motivated by the analysis of physiological time series data in electronic health records, which are sparse, irregularly sampled, and multivariate. We investigate the performance of this architecture on both classification and regression tasks, showing that our approach outperforms a range of baseline and recently proposed models.[1]

## 1 INTRODUCTION

Over the last several years, there has been significant progress in developing specialized models and architectures that can accommodate sparse and irregularly sampled time series as input (Marlin et al., 2012; Li & Marlin, 2015; 2016; Lipton et al., 2016; Futoma et al., 2017; Che et al., 2018a). An irregularly sampled time series is a sequence of samples with irregular intervals between their observation times. Irregularly sampled data are considered to be sparse when the intervals between successive observations are often large. Of particular interest in the supervised learning setting are methods that perform end-to-end learning directly using multivariate sparse and irregularly sampled time series as input without the need for a separate interpolation or imputation step.

In this work, we present a new model architecture for supervised learning with multivariate sparse and irregularly sampled data: Interpolation-Prediction Networks. The architecture is based on the use of several semi-parametric interpolation layers organized into an interpolation network, followed by the application of a prediction network that can leverage any standard deep learning model. In this work, we use GRU networks (Chung et al., 2014) as the prediction network.

The interpolation network allows for information contained in each input time series to contribute to the interpolation of all other time series in the model. The parameters of the interpolation and prediction networks are learned end-to-end via a composite objective function consisting of supervised and unsupervised components. The interpolation network serves the same purpose as the multivariate Gaussian process used in the work of Futoma et al. (2017), but remove the restrictions associated with the need for a positive definite covariance matrix.

Our approach also allows us to compute an explicit multi-timescale representation of the input time series, which we use to isolate information about transients (short duration events) from broader

---

[1]Our implementation is available at : https://github.com/mlds-lab/interp-net

trends. Similar to the work of Lipton et al. (2016) and Che et al. (2018a), our architecture also explicitly leverages a separate information channel related to patterns of observation times. However, our representation uses a semi-parametric intensity function representation of this information that is more closely related to the work of Lasko (2014) on modeling medical event point processes.

Our architecture thus produces three output time series for each input time series: a smooth interpolation modeling broad trends in the input, a short time-scale interpolation modeling transients, and an intensity function modeling local observation frequencies.

This work is motivated by problems in the analysis of electronic health records (EHRs) (Marlin et al., 2012; Lipton et al., 2016; Futoma et al., 2017; Che et al., 2018a). It remains rare for hospital systems to capture dense physiological data streams. Instead, it is common for the physiological time series data in electronic health records to be both sparse and irregularly sampled. The additional issue of the lack of alignment in the observation times across physiological variables is also very common.

We evaluate the proposed architecture on two datasets for both classification and regression tasks. Our approach outperforms a variety of simple baseline models as well as the basic and advanced GRU models introduced by Che et al. (2018a) across several metrics. We also compare our model with to the Gaussian process adapter (Li & Marlin, 2016) and multi-task Gaussian process RNN classifier (Futoma et al., 2017). Further, we perform full ablation testing of the information channels our architecture can produce to assess their impact on classification and regression performance.

## 2 RELATED WORK

The problem of interest in this work is learning supervised machine learning models from sparse and irregularly sampled multivariate time series. As described in the introduction, a sparse and irregularly sampled time series is a sequence of samples with large and irregular intervals between their observation times.

Such data commonly occur in electronic health records, where they can represent a significant problem for both supervised and unsupervised learning methods (Yadav et al., 2018). Sparse and irregularly sampled time series data also occur in a range of other areas with similarly complex observation processes including climate science (Schulz & Stattegger, 1997), ecology (Clark & Bjørnstad, 2004), biology (Ruf, 1999), and astronomy (Scargle, 1982).

A closely related (but distinct) problem is performing supervised learning in the presence of missing data (Batista & Monard, 2003). The primary difference is that the missing data problem is generally defined with respect to a fixed-dimensional feature space (Little & Rubin, 2014). In the irregularly sampled time series problem, observations typically occur in continuous time and there may be no notion of a "normal" or "expected" sampling frequency for some domains.

Methods for dealing with missing data in supervised learning include the pre-application of imputation methods (Sterne et al., 2009), and learning joint models of features and labels (Williams et al., 2005). Joint models can either be learned generatively to optimize the joint likelihood of features and labels, or discriminately to optimize the conditional likelihood of the labels. The problem of irregular sampling can be converted to a missing data problem by discretizing the time axis into non-overlapping intervals. Intervals with no observations are then said to contain missing values.

This is the approach taken to deal with irregular sampling by Marlin et al. (2012) as well as Lipton et al. (2016). This approach forces a choice of discretization interval length. When the intervals are long, there will be less missing data, but there can also be multiple observations in the same interval, which must be accounted for using ad-hoc methods. When the intervals are shorter, most intervals will contain at most one value, but many intervals may be empty. Learning is generally harder as the amount of missing data increases, so choosing a discretization interval length must be dealt with as a hyper-parameter of such a method.

One important feature of missing data problems is the potential for the sequence of observation times to itself be informative (Little & Rubin, 2014). Since the set of missing data indicators is always observed, this information is typically easy to condition on. This technique has been used successfully to improve models in the domain of recommender systems (Salakhutdinov et al., 2007). It was also used by Lipton et al. (2016) to improve performance of their GRU model.

The alternative to pre-discretizing irregularly sampled time series to convert the problem of irregular sampling into the problem of missing data is to construct models with the ability to directly use an irregularly sampled time series as input. The machine learning and statistics literature include several models with this ability. In the probabilistic setting, Gaussian process models have the ability to represent continuous time data via the use of mean and covariance functions (Rasmussen, 2006). These models have non-probabilistic analogues that are similarly defined in terms of kernels.

For example, Lu et al. (2008) present a kernel-based method that can be used to produce a similarity function between two irregularly sampled time series. Li & Marlin (2015) subsequently provided a generalization of this approach to the case of kernels between Gaussian process models. Li & Marlin (2016) showed how the re-parameterization trick (Kingma et al., 2015) could be used to extend these ideas to enable end-to-end training of a deep neural network model (feed-forward, convolutional, or recurrent) stacked on top of a Gaussian process layer. While the basic model of Li & Marlin (2016) was only applied to univariate time series, in follow-up work the model was extended to multivariate time series using a multi-output Gaussian process regression model (Futoma et al., 2017). However, modeling multivariate time series within this framework is quite challenging due to the constraints on the covariance function used in the GP layer. Futoma et al. (2017) deal with this problem using a sum of separable kernel functions (Bonilla et al., 2008), which limit the expressiveness of the model.

An important property of the above models is that they allow for incorporating all of the information from all available time points into a global interpolation model. Variants differ in terms of whether they only leverage the posterior mean when the final supervised problem is solved, or whether the whole posterior is used. A separate line of work has looked at the use of more local interpolation methods while still operating directly over continuous time inputs.

For example, Che et al. (2018a) presented several methods based on gated recurrent unit (GRU) networks (Chung et al., 2014) combined with simple imputation methods including mean imputation and forward filling with past values. Che et al. (2018a) additionally considered an approach that takes as input a sequence consisting of both the observed values and the timestamps at which those values were observed. The previously observed input value is decayed over time toward the overall mean. In another variant the hidden states are similarly decayed toward zero. Yoon et al. (2017) presented another similar approach based on multi-directional RNN which operate across streams in addition to within streams. However, these models are limited to using global information about the structure of the time series via its empirical mean value, and current or past information about observed values. The global structure of the time series is not directly taken into account.

Che et al. (2018b) focus on a similar problem of modeling multi-rate multivariate time series data. This is similar to the problem of interest in that the observations across time series can be unaligned. The difference is that the observations in each time series are uniformly spaced, which is a simpler case. In the case of missing data, they use forward or linear interpolation, which again does not capture the global structure of time series. Similarly, Binkowski et al. (2018) presented an autoregressive framework for regression tasks with irregularly sampled time series data. It is not clear how it can be extended for classification.

The model proposed in this work is similar to that of Li & Marlin (2016) and Futoma et al. (2017) in the sense that it consists of global interpolation layers. The primary difference is that these prior approaches used Gaussian process representations within the interpolation layers. The resulting computations can be expensive and, as noted, the design of covariance functions in the multivariate case can be challenging. By contrast, our proposed model uses semi-parametric, deterministic, feed-forward interpolation layers. These layers do not encode uncertainty, but they do allow for very flexible interpolation both within and across layers.

Also similar to Li & Marlin (2016) and Futoma et al. (2017), the interpolation layers in our architecture produce regularly sampled interpolants that can serve as inputs for arbitrary, unmodified, deep classification and regression networks. This is in contrast to the approach of Che et al. (2018a), where a recurrent network architecture was directly modified, reducing the modularity of the approach. Finally, similar to Lipton et al. (2016), our model includes information about the times at which observations occur. However, instead of pre-discretizing the inputs and viewing this information in terms of a binary observation mask or set of missing data indicators, we directly model the sequence of observation events as a point process in continuous time using a semi-parametric intensity function (Lasko, 2014).

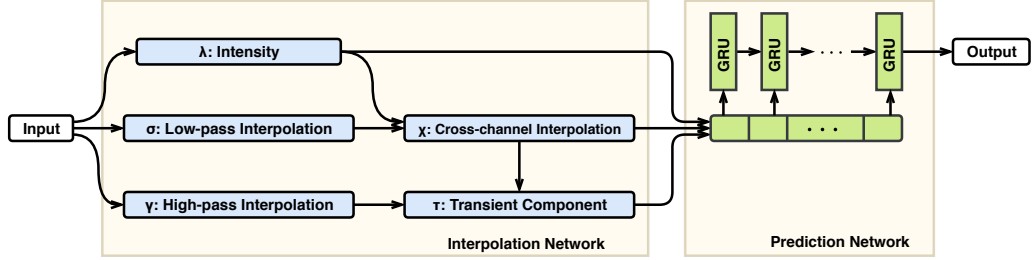

Figure 1: Architecture of the proposed model

# 3 MODEL FRAMEWORK

In this section, we present the proposed modeling framework. We begin by presenting notation, followed by the model architecture and learning criteria.

## 3.1 NOTATION

We let $\mathcal{D} = \{(\mathbf{s}_n, y_n)|n = 1, ..., N\}$ represent a data set containing $N$ data cases. An individual data case consists of a single target value $y_n$ (discrete for classification and real-valued in the case of regression), as well as a $D$-dimensional, sparse and irregularly sampled multivariate time series $\mathbf{s}_n$. Different dimensions $d$ of the multivariate time series can have observations at different times, as well as different total numbers of observations $L_{dn}$. Thus, we represent time series $d$ for data case $n$ as a tuple $\mathbf{s}_{dn} = (\mathbf{t}_{dn}, \mathbf{x}_{dn})$ where $\mathbf{t}_{dn} = [t_{1dn}, ..., t_{L_{dn}dn}]$ is the list of time points at which observations are defined and $\mathbf{x}_{dn} = [x_{1dn}, ..., x_{L_{dn}dn}]$ is the corresponding list of observed values.

## 3.2 MODEL ARCHITECTURE

The overall model architecture consists of two main components: an interpolation network and a prediction network. The interpolation network interpolates the multivariate, sparse, and irregularly sampled input time series against a set of reference time points $\mathbf{r} = [r_1, ..., r_T]$. We assume that all of the time series are defined within a common time interval (for example, the first 24 or 48 hours after admission for MIMIC-III dataset). The $T$ reference time points $r_t$ are chosen to be evenly spaced within that interval. In this work, we propose a two-layer interpolation network with each layer performing a different type of interpolation.

The second component, the prediction network, takes the output of the interpolation network as its input and produces a prediction $\hat{y}_n$ for the target variable. The prediction network can consist of any standard supervised neural network architecture (fully-connected feedforward, convolutional, recurrent, etc). Thus, the architecture is fully modular with respect to the use of different prediction networks. In order to train the interpolation network, the architecture also includes an auto-encoding component to provide an unsupervised learning signal in addition to the supervised learning signal from the prediction network. Figure 1 shows the architecture of the proposed model. We describe the components of the model in detail below.

### 3.2.1 INTERPOLATION NETWORK

We begin by describing the interpolation network. The goal of the interpolation network is to provide a collection of interpolants of each of the $D$ dimensions of an input multivariate time series defined at the $T$ reference time points $\mathbf{r} = [r_1, ..., r_T]$. In this work, we use a total of $C = 3$ outputs for each of the $D$ input time series. The three outputs (discussed in detail below) capture smooth trends, transients, and observation intensity information. We define $f_\theta(\mathbf{r}, \mathbf{s}_n)$ to be the function computing the output $\hat{\mathbf{s}}_n$ of the interpolation network. The output $\hat{\mathbf{s}}_n$ is a fixed-sized array with dimensions $(DC) \times T$ for all inputs $\mathbf{s}_n$.

The first layer in the interpolation network separately performs three semi-parametric univariate transformations for each of the $D$ time series. Each transformation is based on a radial basis function

(RBF) network to accommodate continuous time observations. The transformations are a low-pass (or smooth) interpolation $\boldsymbol{\sigma}_d$, a high-pass (or non-smooth) interpolation $\boldsymbol{\gamma}_d$ and an intensity function $\boldsymbol{\lambda}_d$. These transformations are computed at reference time point $r_k$ for each data case and each input time series $d$ as shown in Equations 1, 2, 3 and 4.[2] The smooth interpolation $\boldsymbol{\sigma}_d$ uses a squared exponential kernel with parameter $\alpha_d$, while the non-smooth interpolation $\boldsymbol{\gamma}_d$ uses a squared exponential kernel with parameter $\kappa\alpha_d$ for $\kappa > 1$.

$$Z(r, \mathbf{t}, \alpha) = \sum_{t \in \mathbf{t}} w(r, t, \alpha), \quad w(r, t, \alpha) = \exp(-\alpha(r - t)^2) \tag{1}$$

$$\lambda_{kd} = h_\theta^\lambda(r_k, \mathbf{t}_d, \mathbf{x}_d) = Z(r_k, \mathbf{t}_d, \alpha_d) \tag{2}$$

$$\sigma_{kd} = h_\theta^\sigma(r_k, \mathbf{t}_d, \mathbf{x}_d) = \frac{1}{Z(r_k, \mathbf{t}_d, \alpha_d)} \sum_{j=1}^{L_{dn}} w(r_k, t_{jd}, \alpha_d)\, x_{jd} \tag{3}$$

$$\gamma_{kd} = h_\theta^\gamma(r_k, \mathbf{t}_d, \mathbf{x}_d) = \frac{1}{Z(r_k, \mathbf{t}_d, \kappa\alpha_d)} \sum_{j=1}^{L_{dn}} w(r_k, t_{jd}, \kappa\alpha_d)\, x_{jd} \tag{4}$$

The second interpolation layer merges information across all $D$ time series at each reference time point by taking into account learnable correlations $\rho_{dd'}$ across all time series. This results in a cross-dimension interpolation $\boldsymbol{\chi}_d$ for each input dimension $d$. We further define a transient component $\boldsymbol{\tau}_d$ for each input dimension $d$ as the difference between the high-pass (or non-smooth) interpolation $\boldsymbol{\gamma}_d$ from the first layer and the smooth cross-dimension interpolation $\boldsymbol{\chi}_d$, as shown in Equation 5.

$$\chi_{kd} = h_\theta^\chi(r_k, \mathbf{s}) = \frac{\sum_{d'} \rho_{dd'}\, \lambda_{kd'}\, \sigma_{kd'}}{\sum_{d'} \lambda_{kd'}}, \qquad \tau_{kd} = h_\theta^\tau(r_k, \mathbf{s}) = \gamma_{kd} - \chi_{kd} \tag{5}$$

In the experiments presented in the next section, we use a total of three interpolation network outputs per dimension $d$ as the input to the prediction network. We use the smooth, cross-channel interpolants $\boldsymbol{\chi}_d$ to capture smooth trends, the transient components $\boldsymbol{\tau}_d$ to capture transients, and the intensity functions $\boldsymbol{\lambda}_d$ to capture information about where observations occur in time.

### 3.2.2 Prediction Network

Following the application of the interpolation network, all $D$ dimensions of the input multivariate time series have been re-represented in terms of $C$ outputs defined on the regularly spaced set of reference time points $r_1, ..., r_T$ (in our experiments, we use $C = 3$ as described above). Again, we refer to the complete set of interpolation network outputs as $\hat{\mathbf{s}}_n = f_\theta(\mathbf{r}, \mathbf{s}_n)$, which can be represented as a matrix of size $(DC) \times T$.

The prediction network must take $\hat{\mathbf{s}}_n$ as input and output a prediction $\hat{y}_n = g_\phi(\hat{\mathbf{s}}_n) = g_\phi(f_\theta(\mathbf{r}, \mathbf{s}_n))$ of the target value $y_n$ for data case $n$. There are many possible choices for this component of the model. For example, the matrix $\hat{\mathbf{s}}_n$ can be converted into a single long vector and provided as input to a standard multi-layer feedforward network. A temporal convolutional model or a recurrent model like a GRU or LSTM can instead be applied to time slices of the matrix $\hat{\mathbf{s}}_n$. In this work, we conduct experiments leveraging a GRU network as the prediction network.

### 3.2.3 Learning

To learn the model parameters, we use a composite objective function consisting of a supervised component and an unsupervised component. This is due to the fact that the supervised component alone is insufficient to learn reasonable parameters for the interpolation network parameters given the amount of available training data. The unsupervised component used corresponds to an autoencoder-like loss function. However, the semi-parametric RBF interpolation layers have the ability to exactly fit the input points by setting the RBF kernel parameters to very large values.

---

[2] We drop the data case index $n$ for brevity in the equations below.

To avoid this solution and force the interpolation layers to learn to properly interpolate the input data, it is necessary to hold out some observed data points $x_{jdn}$ during learning and then to compute the reconstruction loss only for these data points. This is a well-known problem with high-capacity autoencoders, and past work has used similar strategies to avoid the problem of trivially memorizing the input data without learning useful structure.

To implement the autoencoder component of the loss, we introduce a set of masking variables $m_{jdn}$ for each data point $(t_{jdn}, x_{jdn})$. If $m_{jdn} = 1$, then we remove the data point $(t_{jdn}, x_{jdn})$ as an input to the interpolation network, and include the predicted value of this time point when assessing the autoencoder loss. We use the shorthand notation $\mathbf{m}_n \odot \mathbf{s}_n$ to represent the subset of values of $\mathbf{s}_n$ that are masked out, and $(1 - \mathbf{m}_n) \odot \mathbf{s}_n$ to represent the subset of values of $\mathbf{s}_n$ that are not masked out. The value $\hat{x}_{jdn}$ that we predict for a masked input at time point $t_{jdn}$ is the value of the smooth cross-channel interpolant at that time point, calculated based on the un-masked input values: $\hat{x}_{jdn} = h_\theta^\chi(t_{jdn}, (1 - \mathbf{m}_n) \odot \mathbf{s}_n)$.

We can now define the learning objective for the proposed framework. We let $\ell_P$ be the loss for the prediction network (we use cross-entropy loss for classification and squared error for regression). We let $\ell_I$ be the interpolation network autoencoder loss (we use standard squared error). We also include $\ell_2$ regularizers for both the interpolation and prediction networks parameters. $\delta_I$, $\delta_P$, and $\delta_R$ are hyper-parameters that control the trade-off between the components of the objective function.

$$\theta_*, \phi_* = \operatorname*{arg\,min}_{\theta, \phi} \sum_{n=1}^{N} \ell_P(y_n, g_\phi(f_\theta(\mathbf{s}_n))) + \delta_I \|\theta\|_2^2 + \delta_P \|\phi\|_2^2 \tag{6}$$

$$+ \delta_R \sum_{n=1}^{N} \sum_{d=1}^{D} \sum_{j=1}^{L_{dn}} m_{jdn} \ell_I(x_{jdn}, h_\theta^\chi(t_{jdn}, (1 - \mathbf{m}_n) \odot \mathbf{s}_n))$$

## 4 EXPERIMENTS AND RESULTS

In this section, we present experiments based on both classification and regression tasks with sparse and irregularly sampled multivariate time series. In both cases, the input to the prediction network is a sparse and irregularly sampled time series, and the output is a single scalar representing either the predicted class or the regression target variable. We test the model framework on two publicly available real-world datasets: MIMIC-III [3] − a multivariate time series dataset consisting of sparse and irregularly sampled physiological signals collected at Beth Israel Deaconess Medical Center from 2001 to 2012 (Johnson et al., 2016), and UWaveGesture [4] − a univariate time series data set consisting of simple gesture patterns divided into eight categories (Liu et al., 2009). Details of each dataset can be found in the Appendix A.1. We use the MIMIC-III mortality and length of stay prediction tasks as example classification and regression tasks with multivariate time series. We use the UWave gesture classification task for assessing training time and performance relative to univariate baseline models.

### 4.1 BASELINE MODELS

We compare our proposed model to a number of baseline approaches including off-the-shelf classification and regression models learned using basic features, as well as more recent approaches based on customized neural network models.

#### 4.1.1 NON-NEURAL NETWORK BASELINES

For non-neural network baselines, we evaluate Logistic Regression (Hosmer Jr et al., 2013), Support Vector Machines (SVM) (Cortes & Vapnik, 1995), Random Forests (RF) (Breiman, 2001) and AdaBoost (Freund & Schapire, 1997) for the classification task.

For the length of stay prediction task, we apply Linear Regression (Hastie et al., 2001), Support Vector Regression (SVR), AdaBoost Regression (Drucker, 1997) and Random Forest Regression.

---

[3]MIMIC-III is available at `https://mimic.physionet.org/`
[4]UWaveGestureLibraryAll is available at `http://timeseriesclassification.com`.

Standard instances of all of these models require fixed-size feature representations. We use temporal discretization with forward filling to create fixed-size representation in case of missing data and use this representation as feature set for non-neural network baselines.

### 4.1.2 NEURAL NETWORK MODELS

We compare to several existing deep learning baselines built on GRUs using simple interpolation or imputation approaches. In addition, we compare to current state-of-the-art models for mortality prediction including the work of Che et al. (2018a). Their work proposed to handle irregularly sampled and missing data using recurrent neural networks (RNNs) by introducing temporal decays in the input and/or hidden layers. We also evaluate the scalable end-to-end Gaussian process adapter (Li & Marlin, 2016) as well as multi-task Gaussian process RNN classifier (Futoma et al., 2017) for irregularly sampled univariate and multivariate time series classification respectively. This work is discussed in detail in Section 2. The complete set of models that we compare to is as follows:

- **GP-GRU:** End-to-end Gaussian process with GRU as classifier.
- **GRU-M:** Missing observations replaced with the global mean of the variable across the training examples.
- **GRU-F:** Missing values set to last observed measurement within that time series (referred to as forward filling).
- **GRU-S:** Missing values replaced with the global mean. Input is concatenated with masking variable and time interval indicating how long the particular variable is missing.
- **GRU-D:** In order to capture richer information, decay is introduced in the input as well as hidden layer of a GRU. Instead of replacing missing values with the last measurement, missing values are decayed over time towards the empirical mean.
- **GRU-HD:** A variation of GRU-D where decay in only introduced in the hidden layer.

### 4.2 RESULTS

In this section, we present the results of the classification and regression experiments, as well as the results of ablation testing of the internal structure of the interpolation network for the proposed model. We use the UWaveGesture dataset to assess the training time and classification performance relative to the baseline models. We use the standard train and test sets (details are given in appendix A.1). We report the training time taken for convergence along with accuracy on test set.

For MIMIC-III, we create our own dataset (appendix A.1) and report the results of a 5-fold cross validation experiment in terms of the average area under the ROC curve (AUC score), average area under the precision-recall curve (AUPRC score), and average cross-entropy loss for the classification task. For the regression task, we use average median absolute error and average fraction of explained variation (EV) as metrics. We also report the standard deviation over cross validation folds for all metrics.

Training and implementation details can be found in appendix A.2. Figure 2 shows the classification performance on the UWaveGesture dataset. The proposed model and the Gaussian process adapter (Li & Marlin, 2016) significantly outperform the rest of the baselines. However, the proposed model achieves similar performance to the Gaussian process adapter, but with a 50x speed up (note the log scale on the training time axis). On the other hand, the training time of the proposed model is approximately the same order as other GRU-based models, but it achieves much better accuracy.

Table 1 compares the predictive performance of the mortality and length of stay prediction task on MIMIC-III. We note that in highly skewed datasets as is the case of MIMIC-III, AUPRC (Davis & Goadrich, 2006) can give better insights about the classification performance as compared to AUC score. The proposed model consistently achieves the best average score over all the metrics. We note that a paired t-test indicates that the proposed model results in statistically significant improvements over all baseline models ($p < 0.01$) with respect to all the metrics except median absolute error. The version of the proposed model used in this experiment includes all three interpolation network outputs (smooth interpolation, transients, and intensity function).

An ablation study shows that the results on the regression task can be further improved by using only two outputs (transients, and intensity function), achieving statistically significant improvements over

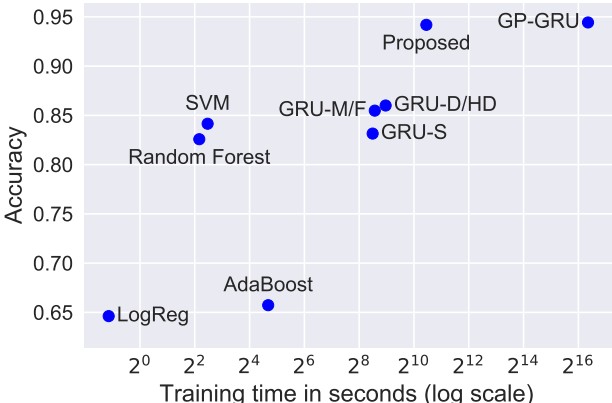

Figure 2: Classification performance on the UWaveGesture dataset. Models with almost same performance are shown with the same dot e.g. (GRU-M, GRU-F ) and (GRU-D, GRU-HD).

Table 1: Performance on Mortality (classification) and Length of stay prediction (regression) tasks on MIMIC-III. Loss: Cross-Entropy Loss, MedAE: Median Absolute Error (in days), EV: Explained variance

| Model | Classification | | | Regression | |
|---|---|---|---|---|---|
| | **AUC** | **AUPRC** | **Loss** | **MedAE** | **EV score** |
| Log/LinReg | $0.772 \pm 0.013$ | $0.303 \pm 0.018$ | $0.240 \pm 0.003$ | $3.528 \pm 0.072$ | $0.043 \pm 0.012$ |
| SVM | $0.671 \pm 0.005$ | $0.300 \pm 0.011$ | $0.260 \pm 0.002$ | $3.523 \pm 0.071$ | $0.042 \pm 0.011$ |
| AdaBoost | $0.829 \pm 0.007$ | $0.345 \pm 0.007$ | $0.663 \pm 0.000$ | $4.517 \pm 0.234$ | $0.100 \pm 0.012$ |
| RF | $0.826 \pm 0.008$ | $0.356 \pm 0.010$ | $0.315 \pm 0.025$ | $3.113 \pm 0.125$ | $0.117 \pm 0.035$ |
| GRU-M | $0.831 \pm 0.007$ | $0.376 \pm 0.022$ | $0.220 \pm 0.004$ | $3.140 \pm 0.196$ | $0.131 \pm 0.044$ |
| GRU-F | $0.821 \pm 0.007$ | $0.360 \pm 0.013$ | $0.224 \pm 0.003$ | $3.064 \pm 0.247$ | $0.126 \pm 0.025$ |
| GRU-S | $0.843 \pm 0.007$ | $0.376 \pm 0.014$ | $0.218 \pm 0.005$ | $2.900 \pm 0.129$ | $0.161 \pm 0.025$ |
| GRU-D | $0.835 \pm 0.013$ | $0.359 \pm 0.025$ | $0.225 \pm 0.009$ | $\mathbf{2.891 \pm 0.103}$ | $0.146 \pm 0.051$ |
| GRU-HD | $0.845 \pm 0.006$ | $0.390 \pm 0.010$ | $0.215 \pm 0.004$ | $\mathbf{2.893 \pm 0.155}$ | $0.158 \pm 0.037$ |
| GP-GRU | $0.847 \pm 0.007$ | $0.377 \pm 0.017$ | $0.215 \pm 0.004$ | $\mathbf{2.847 \pm 0.079}$ | $0.217 \pm 0.020$ |
| **Proposed** | $\mathbf{0.853 \pm 0.007}$ | $\mathbf{0.418 \pm 0.022}$ | $\mathbf{0.210 \pm 0.004}$ | $\mathbf{2.862 \pm 0.166}$ | $\mathbf{0.245 \pm 0.019}$ |

all the baselines. Results for the ablation study are given in Appendix A.3. Finally, we compare the proposed model with multiple baselines on a previous MIMIC-III benchmark dataset (Harutyunyan et al., 2017), which uses a reduced number of cohorts as compared to the one used in our experiments. Appendix A.4 shows the results on this benchmark dataset, where our proposed approach again outperforms prior approaches.

## 5 DISCUSSION AND CONCLUSIONS

In this paper, we have presented a new framework for dealing with the problem of supervised learning in the presence of sparse and irregularly sampled time series. The proposed framework is fully modular. It uses an interpolation network to accommodate the complexity that results from using sparse and irregularly sampled data as supervised learning inputs, followed by the application of a prediction network that operates over the regularly spaced and fully observed, multi-channel output provided by the interpolation network. The proposed approach also addresses some difficulties with prior approaches including the complexity of the Gaussian process interpolation layers used in (Li & Marlin, 2016; Futoma et al., 2017), and the lack of modularity in the approach of Che et al. (2018a). Our framework also introduces novel elements including the use of semi-parametric, feed-forward interpolation layers, and the decomposition of an irregularly sampled input time series into multi-

ple distinct information channels. Our results show statistically significant improvements for both classification and regression tasks over a range of baseline and state-of-the-art methods.

## ACKNOWLEDGEMENTS

This work was supported by the National Science Foundation under Grant No. IIS-1350522.

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

# A  APPENDIX

## A.1  DATASET DESCRIPTIONS

### A.1.1  MIMIC-III DATASET

We evaluate our model framework on the publicly available MIMIC-III dataset (Johnson et al., 2016). MIMIC-III is a de-identified dataset collected at Beth Israel Deaconess Medical Center from 2001 to 2012. It consists of approximately 58,000 hospital admission records. This data set contains sparse and irregularly sampled physiological signals, medications, diagnostic codes, in-hospital mortality, length of stay and more. We focus on predicting in-hospital mortality and length of stay using the first 48 hours of data. We extracted 12 standard physiological variables from each of the 53,211 records obtained after removing hospital admission records with length of stay less than 48 hours. Table 2 shows the features, sampling rates (per hour) and their missingness information computed using the union of all time stamps that exist in any dimension of the input time series.

Table 2: Features extracted from MIMIC III for our experiments

| feature | #Missing | Sampling Rate | feature | #Missing | Sampling Rate |
|---|---|---|---|---|---|
| SpO2 | 31.35% | 0.80 | TGCS | 87.94% | 0.14 |
| HR | 23.23% | 0.90 | CRR | 95.08% | 0.06 |
| RR | 59.48% | 0.48 | UO | 82.47% | 0.20 |
| SBP | 49.76% | 0.59 | FiO2 | 94.82% | 0.06 |
| DBP | 48.73% | 0.60 | Glucose | 91.47% | 0.10 |
| Temp | 83.80% | 0.19 | pH | 96.25% | 0.04 |

Prediction Tasks

In our experiments, each admission record corresponds to one data case $(\mathbf{s_n}, y_n)$. Each data case $n$ consists of a sparse and irregularly sampled time series $\mathbf{s}_n$ with $D = 12$ dimensions. Each dimension $d$ of $\mathbf{s}_n$ corresponds to one of the 12 vital sign time series mentioned above. In the case of classification, $y_n$ is a binary indicator where $y_n = 1$ indicates that the patient died at any point within the hospital stay following the first 48 hours and $y_n = 0$ indicates that the patient was discharged at any point after the first 48 hours. There are 4310 (8.1%) patients with a $y_n = 1$ mortality label. The complete data set is $\mathcal{D} = \{(\mathbf{s_n}, y_n) | n = 1, ..., N\}$, and there are $N = 53,211$

data cases. The goal in the classification task is to learn a classification function $g$ of the form $\hat{y}_n \leftarrow g(\mathbf{s}_n)$ where $\hat{y}_n$ is a discrete value.

In the case of regression, $y_n$ is a real-valued regression target corresponding to the length of stay. Since the data set includes some very long stay durations, we let $y_n$ represent the log of the length of stay in days for all models. We convert back from the log number of days to the number of days when reporting results. The complete data set is again $\mathcal{D} = \{(\mathbf{s_n}, y_n) | n = 1, ..., N\}$ with $N = 53,211$ data cases (we again require 48 hours worth of data). The goal in the regression task is to learn a regression function $g$ of the form $\hat{y}_n \leftarrow g(\mathbf{s}_n)$ where $\hat{y}_n$ is a continuous value.

### A.1.2 UWave Dataset

UWave dataset is an univariate time series data consisting of simple gesture patterns divided into eight categories. The dataset has been split into 3582 train and 896 test instances. Out of the training data, 30% is used for validation. Each time series contains 945 observations. We follow the same data preparation method as in Li & Marlin (2016) where we randomly sample 10% of the observations points from each time series to create a sparse and irregularly sampled data.

### A.2 Implementation Details

### A.2.1 Proposed Model

The model is learned using the Adam optimization method in TensorFlow with gradients provided via automatic differentiation. However, the actual multivariate time series representation used during learning is based on the union of all time stamps that exist in any dimension of the input time series. Undefined observations are represented as zeros and a separate missing data mask is used to keep track of which time series have observations at each time point. Equations 1 to 5 are modified such that data that are not available are not taken into account at all. This implementation is exactly equivalent to the computations described, but supports parallel computation across all dimensions of the time series for a given data case.

Finally, we note that the learning problem can be solved using a doubly stochastic gradient based on the use of mini batches combined with re-sampling the artificial missing data masks used in the interpolation loss. In practice, we randomly select 20% of the observed data points to hold out from every input time series.

For the time series missing entirely, our interpolation network assigns the starting point (time t=0) value of the time series to the global mean before applying the two-layer interpolation network. In such cases, the first interpolation layer just outputs the global mean for that channel, but the second interpolation layer performs a more meaningful interpolation using the learned correlations from other channels.

### A.2.2 Baselines

The Logistic Regression model is trained with cross entropy loss with regularization strength set to 1. The support vector classifier is used with a RBF kernel and trained to minimize the soft margin loss. We use the cross entropy loss on the validation set to select the optimal number of estimators in case of Adaboost and Random Forest. Similar to the classification setting, the optimal number of estimators for regression task in Adaboost and Random Forest is chosen on the basis of squared error on validation set.

### MIMIC-III Dataset

We evaluate all models using a five-fold cross-validation estimate of generalization performance. In the classification setting, all the deep learning baselines are trained to minimize the cross entropy loss while the proposed model uses a composite loss consisting of cross-entropy loss and interpolation loss (with $\delta_R = 1$) as described in section 3.2.3. In the case of the regression task, all baseline models are trained to minimize squared error and the proposed model is again trained with a composite loss consisting of squared error and interpolation loss.

We follow the multi-task Gaussian process implementation given by Futoma et al. (2017) and treat the number of hidden units and hidden layers as hyper-parameters. For all of the GRU-based models, we use the already specified parameters (Che et al., 2018a). The models are learned using the Adam optimization. Early stopping is used on a validation set sub-sampled from the training folds. In the classification case, the final outputs of the GRU hidden units are used in a logistic layer that predicts the class. In the regression case, the final outputs of the GRU hidden units are used as input for a dense hidden layer with 50 units, followed by a linear output layer.

### UWAVE DATASET

We independently tune the hyper-parameters of each baseline method. For GRU-based methods, hidden units are searched over the range $\{2^5, 2^6, \cdots, 2^{11}\}$. Learning is done in same way as described above. We evaluate all the baseline models on the test set and compare the training time and accuracy. For the Gaussian process model, we use the squared exponential covariance function. We use the same number of inducing points for both the Gaussian process and the proposed model. The Gaussian process model is jointly trained with the GRU using stochastic gradient descent with Nesterov momentum. We apply early stopping based on the validation set.

### A.3    ADDITIONAL EXPERIMENTS

In this section, we address the question of the relative information content of the different outputs produced by the interpolation network used in the proposed model for MIMIC-III dataset. Recall that for each of the $D = 12$ vital sign time series, the interpolation network produces three outputs: a smooth interpolation output (SI), a non-smooth or transient output (T), and an intensity function (I). The above results use all three of these outputs.

To assess the impact of each of the interpolation network outputs, we conduct a set of ablation experiments where we consider using all sub-sets of outputs for both the classification task and for the regression task.

Table 3 shows the results from five-fold cross validation mortality and length of stay prediction experiments. When using each output individually, smooth interpolation (SI) provides the best performance in terms of classification. Interestingly, the intensity output is the best single information source for the regression task and provides at least slightly better mean performance than any of the baseline methods shown in Table 1. Also interesting is the fact that the transients output performs significantly worse when used alone than either the smooth interpolation or the intensity outputs in the classification task.

Table 3: Performance of all subsets of the interpolation network outputs on Mortality (classification) and Length of stay prediction (regression) tasks. SI: Smooth Interpolation, I: Intensity, T: Transients, Loss: Cross-Entropy Loss, MedAE: Median Absolute Error, EV: Explained variance

| Model | Classification | | | Regression | |
|-------|------|-------|------|-------|----------|
|  | **AUC** | **AUPRC** | **Loss** | **MedAE** | **EV score** |
| SI, T, I | **0.853 ± 0.007** | **0.418 ± 0.022** | **0.210 ± 0.004** | 2.862 ± 0.166 | 0.245 ± 0.019 |
| SI, I | 0.852 ± 0.005 | 0.408 ± 0.017 | 0.210 ± 0.004 | 2.745 ± 0.062 | 0.224 ± 0.010 |
| SI, T | 0.820 ± 0.008 | 0.355 ± 0.024 | 0.226 ± 0.005 | 2.911 ± 0.073 | 0.182 ± 0.009 |
| SI | 0.816 ± 0.009 | 0.354 ± 0.018 | 0.226 ± 0.005 | 3.035 ± 0.063 | 0.183 ± 0.016 |
| I | 0.786 ± 0.010 | 0.250 ± 0.012 | 0.241 ± 0.003 | **2.697 ± 0.072** | 0.251 ± 0.009 |
| I, T | 0.755 ± 0.012 | 0.236 ± 0.014 | 0.272 ± 0.010 | 2.738 ± 0.101 | **0.290 ± 0.010** |
| T | 0.705 ± 0.009 | 0.192 ± 0.008 | 0.281 ± 0.004 | 2.995 ± 0.130 | 0.207 ± 0.024 |

When considering combinations of interpolation network components, we can see that the best performance is obtained when all three outputs are used simultaneously in classification tasks. For the regression task, the intensity output provides better performance in terms of median absolute error while a combination of intensity and transients output provide better explained variance score. However, the use of the transients output contributes almost no improvement in the case of the AUC and cross entropy loss for classification relative to using only smooth interpolation and intensity. Inter-

estingly, in the classification case, there is a significant boost in performance by combining smooth interpolation and intensity relative to using either output on its own. In the regression setting, smooth interpolation appears to carry little information.

## A.4 BENCHMARK MIMIC-III DATASET

In this section, we compare the performance of the proposed model on a previous MIMIC-III benchmark dataset (Harutyunyan et al., 2017). This dataset only consists of patients with age $> 18$. Again, we focus on predicting in-hospital mortality using the first 48 hours of data. This yields training and test sets of size 17,903 and 3,236 records respectively.

We compare the proposed model to multiple baselines from Harutyunyan et al. (2017). In all the baselines, the sparse and irregularly sampled time-series data has been discretized into 1-hour intervals. If there are multiple observations in an interval, the mean or last observation is assigned to that interval, depending on the baseline method. Similarly, if an interval contains no observations, the mean or forward filling approach is used to assign a value depending on the baseline method. We compare with a logistic regression model and a standard LSTM network. In the multitask setting, multiple tasks are predicted jointly. Unlike the standard LSTM network where the output/hidden-state from the last time step is used for prediction, we provide supervision to the model at each time step. In this experiment, we use an LSTM as the prediction network in the proposed model to match the baselines.

Table 4: Classification performance for in-hospital mortality prediction task on benchmark dataset

| Model | AUC score | AUPRC score |
|---|---|---|
| Logistic Regression | 0.8485 | 0.4744 |
| LSTM | 0.8547 | 0.4848 |
| LSTM + Deep Supervision | 0.8558 | 0.4928 |
| Multitask LSTM | 0.8607 | 0.4933 |
| **Interpolation Network + LSTM** | **0.8610** | **0.5370** |

