# OpenReview forum: "Interpolation-Prediction Networks for Irregularly Sampled Time Series"
_ICLR.cc/2019/Conference_

### Official Review · AnonReviewer3 · 2018-11-02
**Interesting but still immature solutions to a critical issues in EHRs**

**Rating:** 6
**Confidence:** 4

**Review:**

In the submitted manuscript, the authors introduce a novel deep learning architecture to solve the  problem of supervised learning with sparse and irregularly sampled multivariate time series, with a specific interest in EHRs. The architecture is based on the use of a semi-parametric interpolation network followed by the application of a prediction network, and it is tested on two classification/regression tasks.

The manuscript is interesting and well written: the problem is properly located into context with extensive bibliography, the method is sufficiently detailed and the experimental comparative section is rich and supportive of the authors’ claim. However, there are a couple of issues that need to be discussed:

	▪	the reported performances represent only a limited improvement over the comparing baselines, indicating that the proposed model is promising but it is still immature
	▪	the model is sharing many characteristics with (referenced) published methods, which the proposed algorithm is a smart combination of - thus, overall, the novelty of the introduced method is somewhat limited.


#########

After considering the proposed improvements, I decided to raise my mark to 6. Thanks for the good job done!

---

> ### Author Response · Authors · 2018-11-16
> **Response to Reviewer 3**
>
> Thank you for your comments. We address the issues below:
>
> Q: the reported performances represent only a limited improvement ...
> A:  In Table 1 (UWave dataset), the proposed model achieves similar accuracy to the Gaussian Process (GP) baseline while running 50x faster. At the same time, it outperforms the other strong deep learning baselines. Our model allows for incorporating all of the information from all available time points into a global interpolation model just like GP but removes the restrictions associated with the need for a positive definite covariance matrix and at the same time reduces the computational complexity.
>
> In Table 2 (MIMIC-III dataset), our model achieves statistically significant improvements over the baseline models (p<0.01) with respect to all the metrics except median absolute error. We show that the performance on the regression task can be further improved using only two interpolants (Appendix A.3). We would also like to note that AUPRC (Davis & Goadrich, 2006) is a better metric for a highly imbalanced dataset which is the case here. When considering AUPRC, the difference between the performance of the proposed model with respect to the other baselines increases. Similarly for the regression task, even though the median absolute error is similar to the baselines the explained variance score shows large improvements compared to the baselines.
>
> Thus,  we feel that the improved accuracy relative to the existing GRU-based methods on MIMIC-III coupled with the increased modeling flexibility and significant speed-ups relative to the GP-GRU are important contributions.
>
>
> Q: the model is sharing many characteristics with (referenced) published methods ....
> A: The proposed model is designed to allow the flexible selection of prediction networks, which is characteristic that it shares with the prior GP-based methods. Here, the primary contribution of our approach is a highly significant reduction in the compute time relative to using GP-based methods, which makes the method much more suitable for practical use. In addition, our approach to decomposing the continuous time data to directly expose smooth trends and transient components is absent from prior GP-based methods. Relative to prior neural network based approaches (the GRU-* family), our method focuses on enabling global interpolation and direct use of continuous time data with no ad-hoc decisions about how to assign values to discrete time intervals. These are significant differences relative to the prior approaches, particularly in terms of the interpolation process. Indeed, these differences between global learned interpolation and local imputation directly account for the improved performance of our approach over the GRU-* family of methods.

---

### Official Review · AnonReviewer2 · 2018-11-03
**Possibly, simple yet effective solution to handle time series data with  missing values**

**Rating:** 6
**Confidence:** 4

**Review:**

Summary:
The authors propose a framework for making predictions on a sparse, irregularly sampled time-series data. The proposed model consists of an interpolation module, and the prediction module, where the interpolation module models the missing values in using three outputs: smooth interpolation, non-smooth interpolation, and intensity. The authors test the proposed method on two different datasets (MIMIC-III and UWave), although only one of the datasets are multi-variate. The proposed method shows comparable training time to other GRU variants, and outperforms all baseline models for mortality prediction and length-of-stay prediction.

Pros:
- Possibly, simple yet effective solution to handle time series data with missing values.
- I appreciate the thorough survey of the related works.

Issues:
- My biggest concern is that the authors spend some time to address the disadvantage of discretizing the timeline when modeling missing values (5th paragraph of section 2) and emphasize how their method does not have such limitation. But it seems that, when using the proposed method, the user still needs to pre-define evenly spaced reference points r_1, r_2, ..., r_T. So there is still this dilemma how dense you want the reference points to be. And I couldn't find the values used for the reference points in the experiments section. It's quite possible that one of the baselines can outperform the proposed method with different reference points, given that the evaluation scores overlap with each other wrt standard deviation ranges.
- Method description in section 3.2.1 is quite confusing. I could follow until Eq.2, but afterwards, the first interpolants (x^{21}) and the second interpolants (x^{12}) become very confusing. It would have been helpful if the authors explicitly described what the interpolation channel 'c' was before talking about the interpolants.
- "taking into account learned correlations" in page 5: I suggest changing that to "taking into account learnable/trainable correlations" since "learned correlations" gives the impression that the correlations were already learned prior to training the model.
- Can the authors test the proposed method on logistic regression (LR) and multi-layer perceptron (MLP)? It would be interesting to see if the proposed method improves the performance of LR and MLP.

After considering the author feedback and their effort to address my concerns, I've decided to raise my rating to 6. Thank you for the hard work.

---

> ### Author Response · Authors · 2018-11-18
> **Response to Reviewer 2**
>
> Thank you for your helpful comments. We address the issues below:
>
> Q: My biggest concern is that the authors spend some time to address the disadvantage...
> A: The reviewer is correct to point out that the difference between discretizing an input time series into fixed-length intervals and interpolating the same data against regularly space interpolating points can be quite subtle. As noted in the paper, perhaps the most important distinction between these approaches is how they deal with the case of assigning values to the desired discrete time points. Temporal discretization of irregularly sampled continuous time data is often viewed as a pre-processing operation performed outside of the model being learned. As a result, ad-hoc methods for assigning values to intervals are commonly used (such as the average of the observations that fall in an interval). The interpolation process instead conditions on the raw continuous time data and the process of assigning values to interpolation points becomes learnable from the raw data itself as part of the model being estimated (as it is in our approach). This avoids the need to make ad-hoc decisions outside of the model being learned. Another difference is how empty intervals are treated in the temporal discretization approach. In non-probabilistic models, empty intervals are typically treated as missing data, with many existing methods using basic imputation to fill the missing values, again as a pre-processing step that occurs outside of the primary model and often with no learnable parameters (such as forward filling). The interpolation approach completely avoids the artificial creation of missing data, and instead presents what is in a strong sense the true problem with irregularly sampled continuous time data: deciding what value to assign at an interpolation point that may be far from any continuous time observations. Again, this problem can be solved in an end-to-end fashion using learning when posed in this form (as it is in our approach).
>
> Q:  Method description in section 3.2.1 is quite confusing...
> A: We have added a footnote to explain the subscripts and superscripts more clearly. In the paragraphs below equation 2, we explain what interpolants are used in our experiment. We use a smooth interpolant (x^{21}: output of the second layer of interpolation network), a transient component (obtained by subtracting the smooth component x^{21} from first layer interpolant with higher bandwidth x^{12}) and an intensity component (i^1 obtained from the 1st layer).
>
> Q: "taking into account learned correlations" in page 5...
> A: We have updated the paper with your suggestion.
>
> Q: Can the authors test the proposed method on logistic regression (LR) and multi-layer perceptron (MLP)?
> A:  Our proposed framework is highly flexible and can be used with any differentiable network on top of the interpolation layers. As request, we replaced the GRU prediction network with a simpler Logistic Regression network and a fully connected feed-forward network (MLP). We report the results below:
>
> Model               UWave(Accuracy)        MIMIC-III (AUC on mortality classification task)
> IpN + LR .                   0.878                     0.78 +/- 0.010
> IpN + MLP                  0.877                    0.77 +/- 0.010
> IpN + GRU                  0.942                    0.85 +/- 0.007
> *IpN: Proposed Interpolation Network
> Increasing the size of the hidden layer in MLP leads to overparameterization and thus reduces the performance.

---

### Official Review · AnonReviewer1 · 2018-11-07
**Refreshingly simple approach to irregular data but limited novelty, flawed writing, uninspiring results**

**Rating:** 6
**Confidence:** 4

**Review:**

I have mixed feelings about this submission, and as such, I look forward to discussing it with both the authors and my fellow reviewers. In short, I like the simplicity of the idea, but I am uncertain about the degree to which it satisfies ICLR's novelty criterion ("present substantively new ideas or explore an underexplored or highly novel question"); I do feel confident that some ICLR readers would (perhaps unfairly) describe this approach as "obvious." The paper's presentation suffers, and it fails to communicate essential details clearly. Finally, for folks familiar with healthcare data and MIMIC-III specifically, the results are underwhelming: yes, the proposed approach beats (the authors' own implementations of) baselines, but it underperforms other published results on the MIMIC-III 48-hour mortality task ([1][2] report AUCs of 0.87 or higher). As such, I am assigning the paper a "weak accept" to communicate my ambivalence and reserve the right to adjust it up or down after discussion.

SUMMARY

This paper proposes an "interpolation layer" to resample irregularly sampled time series before feeding them into a neural net architecture. The interpolation layer consists of parametric kernels, e.g., radial basis functions, configured to estimate the values of input time series at reference time points based on univariate temporal and then multivariate correlations. The outputs include smooth and transient interpolated values (controlled by kernel bandwidth) and counts (referred to as intensity) at each reference point. As far as I understand, this model can be trained end-to-end. The paper also proposes a simple strategy for combatting overfitting (add an autoencoder and reconstruction error term to the objective in combination with a heuristic in which some points are masked as inputs and must be interpolated from non-masked points). In experiments on two data sets (UWaveGesture and a medical data set) and two tasks (classification and regression) this approach outperforms the main competing approaches [3][4][5][6] in most contexts.

Below I provide a list of strengths, weaknesses, and general questions or feedback.

STRENGTHS

- I applaud the simplicity of the idea: this much simpler framework leverages many of the intuitions behind the GP adapter framework (GP-GRU) [4][5] with comparable performance and appears to train orders of magnitude faster (caveat: on one data set and task)
- It likewise outperforms both commonly used preprocessing (GRU-F) [2][3] and the much more complicated neural net architecture (GRU-HD) from [6] (across two datasets and tasks)
- The simplicity of this approach probably lends itself to additional customization and innovation
- The literature review seems quite thorough and does an especially nice job of covering recent work on RNNs for multivariate time series and irregular sampling or missing values
- The experiments are thorough and well-designed overall. The authors use two data sets and two tasks (classification and regression). More data sets and tasks is always nice, but even two is pretty laudable (many authors might settle on just one given the experimental and computational effort required for these experiments). They include and beat or outperform two baselines that can justifiably be called state-of-the-art (GP-GRU and GRU-HD).

I think a relatively safe takeaway is that for irregularly sampled data, this approach is is preferable to both heuristic preprocessing and more complex models. That seems like a not insignificant finding in empirical machine learning for messy time series data.

WEAKNESSES

- Section 3 is possibly the most critical section (since it describes the contribution) but is hard to follow: I don't envy the authors the task of explaining a variable with two superscripts and three subscripts (Equation 1), but it IS their paper, so it's on them to do it. See feedback section for other examples.
- Although I consider the related work well done, I can't help but wonder if there isn't older work on RBFs, etc., that might have been missed (I mostly want to encourage the authors to look once more and then come back and tell me I'm wrong).
- The MIMIC-III experiments omit the GP-GRU model, which weakens the results by leaving the reader to imagine how it might compare (I would expect it to outperform the proposed approach by an even wider margin than it did for UWave).
- I am sympathetic to the idea of fixing certain architectural choices, e.g., layers and units in the GRU and number of inducing points, across all models because it (a) gives the appearance of a "fair comparison" and (b) reduces burden of effort, but I do not agree that it yields a truly fair comparison. The GRU-* model performance on UWave is suspiciously bad, suggesting severe overfitting and the possibility that the models are overparameterized. It leaves the reader wondering if the architectural choices happen to be optimal for the proposed model only (whether by accident or design). A truly fair comparison requires independently tuning hyperparameters for each model.
- Although the proposed approach outperforms baselines in these experiments, the overall results are underwhelming in the wider context of recent work using MIMIC-III. Multiple publications have reported AUCS of 0.87 [1][2] or higher for 48-hour risk of mortality (it is difficult to compare the LOS results since different papers use different units). Of course, the experiments use different cohorts and variables so they're not directly comparable, but it nonetheless diminishes the potential impact of the results presented here.

FEEDBACK AND QUESTIONS

- I had to read 3.2.1 multiple times to understand the relationships between the different "layers" in the interpolator, and I'm still not sure what the relationship is between the smooth and transient kernels or exactly how the intensity values are estimated (are they just windowed counts or weighted sums?).
- I'm also not 100% clear on (a) which parameters (if any) in the interpolator are optimized during end-to-end learning and which are just fixed or tuned as hyperparameters. This should be stated clearly and even better, I'd recommend writing down the gradient update rules for the interpolator parameters (you can put them in the appendix).
- Since the model uses global structure for interpolation and requires pre-specifying the number of inducing points, could it be used to make continuous predictions (and how?), e.g., forecast mortality at each hour?
- On a related note, if the number of inducing points is pre-specified, can the model be applied to sequences of different length?
- How does performance depend on choice of number of inducing points?
- How does the proposed approach handle time series that are missing entirely, e.g., if no pH values are measured?
- What does Table 3 in the appendix mean by "missingness?" Given that the paper is concerned with irregular sampling (not missing data), I would expect statistics on sampling rates, not missingness...
- Why derive your own MIMIC-III subset and tasks rather than use one of several pre-existing benchmarks (both of which include more variables and tasks) [1][2]?
- FYI: the Che, et al., 2016, paper on missing values [6] has been published in JBIO, so you should cite that version.

REFERENCES

[1] Purushotham, et al. "Benchmark of Deep Learning Models on Large Healthcare MIMIC Datasets." arXiv preprint arXiv:1710.08531 (2017)
[2] Harutyunyan, et al. "Multitask learning and benchmarking with clinical time series data." arXiv preprint arXiv:1703.07771 (2017)
[3] Lipton, Kale, and Wetzel, 2016
[4] Li and Marlin, 2016.
[5] Futoma, et al., 2017.
[6] Che, et al., 2016. <-- new JBIO 2018 version!

---

> ### Author Response · Authors · 2018-11-17
> **Thank you for your insightful and detailed comments.**
>
> Thank you for your insightful and detailed comments. We address your concerns below:
>
> Q: The MIMIC-III experiments omit the GP-GRU model...
> A: We omit the GP-GRU model from MIMIC-III experiments because the available reference implementation used in the original work is restricted to the case of modeling univariate time series. Extending it to model multivariate time series is highly non-trivial and the resulting method would have a computational complexity gap relative to the proposed approach that is at least as high as what we observe in the univariate case depending on the covariance functions used.
>
> Q: I am sympathetic to the idea of fixing certain architectural choices...
> A: We have updated the results on UWave dataset after tuning the hyperparameters independently. Interestingly, the GRU-* models achieve improved performance when using a much larger hidden representation size. Our approach still significantly outperforms this collection of models. For the MIMIC-III baselines, our results use the GRU hyper-parameters as specified in the original work (Che at el, 2018a).
>
> Q: Although I consider the related work well done...
> A: The closest related work is the Gaussian Process with RBF kernel (aka squared-exponential kernel) which we have already compared to the proposed model. We couldn't find any other older work on RBFs for modeling irregularly sampled time series.
>
> Q:  I had to read 3.2.1 multiple times to understand the relationships between the different "layers" in the interpolator...
> A: The first interpolation layer performs a semi-parametric univariate interpolation for each of the D time series separately while the second layer merges information from across all of the D time series at each reference time point by taking into account the correlations among the time series. The difference between transient and smooth interpolation is the bandwidth parameters. For the transient component, we use a higher bandwidth parameter to capture the sudden peaks and drops (5th paragraph, Section 3.2.1). Also, to further minimize the redundancy, we subtract the smooth interplant from the transient component. We use an intensity function to retain information about where observations occur. Intensity values are computed using Equation 2. It is the sum of the weights (closeness to a given point) of all observations on a given reference point.
>
> Q:  I'm also not 100% clear on (a) which parameters...
> A: For the interpolation network, the parameters we learn are $alpha_{d1}$ and $rho_{dd’}$ (sec 3.2.1 para 4) for a given number of reference points as described in Section 3.2 (1st paragraph).
>
> Q: On a related note, if the number of inducing points is pre-specified...
> A: Our model can be applied to sequences of different length. The interpolation network interpolates the multivariate, sparse, and irregularly sampled input time series against a set of evenly-spaced reference time points for the prediction network. No preprocessing is required for applying our model to sequences of varying length. Different length sequences can be two types.
> a) Our model handles the varying length sequences defined over the same time period (e.g. 24 hrs or 48 hrs in case of MIMIC-III) by interpolating it over a fixed set of reference time points. In our experiments on MIMIC-III dataset, each dimension of the input time series is of varying length.
> b) In case the varying length sequences cover different time periods we can define the number of reference points per unit time. This leads to a higher number of reference points for higher time periods while keeping a fixed spacing between the reference points. Such regularly-spaced varying length sequences cases can be easily handled by using a recurrent model as a prediction network. For example, let’s assume one data case has observations recorded over 24 hour period while another data case covers 48 hour period. If we specify the number of reference points to be 1 per hour, our interpolation network would lead to an output of length 24 and 48. Since these outputs have fixed spacing between successive points, they can be easily modeled using RNN type model even though they are of varying length.
>
> Q: How does performance depend on the choice of the number of inducing points?
> A: The performance increases with the number of inducing points up to a certain point and remains there if the network is not overparameterized (too many inducing points).
>
> Q: How does the proposed approach handle time series that are missing entirely...
> A:  For the time series missing entirely, the first interpolation layer just outputs the global mean for that channel, but the second interpolation layer performs a more meaningful interpolation using the learned correlations from other channels.

---

> > ### Author Response · Authors · 2018-11-18
> > **Continued..**
> >
> >
> > Q: Could it be used to make continuous predictions (and how?)...
> > A: Since our interpolations model outputs a fixed length representation of the input time series over a given set of reference points, we can use it to make continuous predictions. There could be two possible cases:
> > a) Continuous prediction as the data come: This is a trivial case. We can fix the look-back window and apply our model on a rolling basis.
> >
> > b) Continuous predictions after observing all the data: For the experiments in the paper, our model uses the observations over given time points to interpolate over a fixed set of reference points over the given time period. For continuous predictions, we would require the model to extrapolate over the future time points in order to make predictions. We can achieve this by using our interpolation network recursively (one step at a time) to extrapolate over future time points. For example, since we currently use 48 hours of data to make mortality predictions our reference points are evenly-spaced between 48-hour interval. For continuous predictions, we can use multiple layers of our interpolation network where 1st layer interpolates over the given 48 hour period while the 2nd layer performs extrapolations over the next one hour or so using the outputs of 1st layer and so on. In order to make the rolling predictions, the look back window must be kept fixed (e.g. last 48-hour data). Finally, we can use the output of final interpolation layer to make the predictions using any standard prediction network.
> >
> > Q: What does Table 3 in the appendix mean by "missingness?" ...
> > A: Missing information in Table 2 is computed using the union of all time stamps that exist in any dimension of the input time series. We show this because our baseline methods convert the problem of irregular sampling into a missing data problem. We have also updated this table with sampling rates.
> >
> > Q: Why derive your own MIMIC-III subset...
> > A: The experiments in [1, 2] use a reduced number of cohorts as compared to the one used in our experiments. All the experiments in [1, 2] are done on cohorts with age>18, thus ignoring the children and newborns, which represent approximately 20% of the data. Further preprocessing significantly reduces the number of data cases to 22k[2]. On contrary, we include all the data cases (~ 52k) with the length of stay more than 48 hours.
> >
> > Q: FYI: the Che, et al., 2016, paper on missing value...
> > A: We have updated the citation. Thanks for reminding.

---

> > > ### Comment · AnonReviewer1 · 2018-11-18
> > > **Regarding comparison vs. multivariate GP-GRU**
> > >
> > > Thanks for the switch response!
> > >
> > > > We omit the GP-GRU model from MIMIC-III experiments...
> > >
> > > This is a reasonable response. I happen to think that the burden of reproducibility is on the previous publication, i.e., if they expect subsequent research to compare against their method in a given setting (here, multivariate time series), then they should provide a publicly accessible, easy-to-use, reliable implementation. Thus, I think it would be unfair to punish you for not comparing your approach to a multivariate version of the GP-GRU. However, we need to add two caveats:
> > >
> > > (1) Nonetheless, the absence of a multivariate GP-GRU baseline weakens your paper. I don't think that necessarily requires us to reject your submission from ICLR, but it certainly limits your contribution. There is an open question about how you'd compare in the multivariate regime and plenty of reason to think the GP-GRU would capture multivariate correlations better.
> > >
> > > (2) You're not completely off the hook! Futoma's version of the MGP-RNN, which you cite and which is closely related to the Li and Marlin univariate baseline, IS available on github: https://github.com/jfutoma/MGP-RNN. Further, it looks pretty usable, and Futoma himself is pretty responsive and would be willing to assist you in performing a comparison.
> > >
> > > Returning to my original philosophical point, regarding this footnote from your paper:
> > >
> > > > We plan to share all data extraction and model code on Github.
> > >
> > > We all know that for every ten papers that include a statement like that in a submission, like 1-2 actually publish their code after acceptance. How close are you to ACTUALLY publishing your code? No need to provide a link or anything (we don't want to violate double blind) -- I'm just looking for a forthright reply!

---

> > > > ### Author Response · Authors · 2018-11-28
> > > > **Comparison with MGP-RNN**
> > > >
> > > > Thanks for the quick response. We ran the experiments with MGP-RNN (Futoma's version) on our dataset. In the table below, we report the results from the 5-fold cross validation in terms of the average area under the ROC curve (AUC score) and average area under the precision-recall curve (AUPRC score). We also report the standard deviation over cross-validation folds.
> > > >
> > > > Model                         AUC                        AUPRC
> > > > MGP-RNN          0.847 +/- 0.007       0.377 +/- 0.017
> > > > GRU-HD             0.845 +/- 0.006       0.390 +/- 0.010
> > > > Proposed           0.853 +/- 0.007       0.418 +/- 0.022
> > > >
> > > > The proposed model results in statistically significant improvements over the baseline models (p < 0.01) with respect to both the metrics. The proposed approach also addresses some difficulties with prior approaches including the complexity of the Gaussian process interpolation layers used in Li & Marlin, 2016 and Futoma et al. 2017, and the lack of modularity in the approach of Che et al. (2018a). Our framework also introduces novel elements including the use of semi-parametric, feed-forward interpolation layers, and the decomposition of an irregularly sampled input time series into multiple distinct information channels.
> > > >
> > > > Publishing the code: We actually plan to release all the code (including the dataset) right after the decision deadline.

---

> > > > > ### Comment · AnonReviewer1 · 2018-11-28
> > > > > **Appears to outperform Futoma, et al.**
> > > > >
> > > > > Great work on those experiments, and kudos for acting on my feedback so quickly. If you're still allowed to upload an updated manuscript, please add these results. I am leaning toward acceptance, but I need to confer with the other reviewers and the chair before I revise my score.

---

> > > > > > ### Author Response · Authors · 2018-12-02
> > > > > > **Thank You!**
> > > > > >
> > > > > > We thank you for your helpful reviews once again. We will add the comparison results with MGP-RNN in the final paper.

---

> > > ### Comment · AnonReviewer1 · 2018-11-18
> > > **Other miscellaneous comments**
> > >
> > > Great response overall. In the interest of conserving space and time, you can assume that I am satisfied with any answer I don't directly reply to!
> > >
> > > > The first interpolation layer performs...
> > >
> > > Nice succinct explanation with plain language. I would recommend putting THAT in your paper. ;)
> > >
> > > > For the time series missing entirely, the first interpolation layer just outputs the global mean for that channel...
> > >
> > > Can you further explain "global" here? Is it global across all data points in the data set, across measurements at that particular time, etc.? Also, this is not immediately obvious from the description in Section 3.2.1. The summations in Equation (1) are over only those measurements in an individual record, not across all records, so where does the global mean get computed and how does it come into play? Is substituting the global mean just an ad hoc post-processing step?
> > >
> > > > Continuous predictions after observing all the data:...
> > >
> > > > Missing information in Table 2...our baseline methods convert the problem of irregular sampling into a missing data problem...
> > >
> > > Makes sense. Once again, put this plain language explanation in your paper!
> > >
> > > > The experiments in [1, 2] use a reduced number of cohorts...
> > >
> > > I don't want to belabor this too much: ultimately, it's your choice! However, the community is embracing the notion of shared benchmarks to help accelerate progress and promote reproducibility [2]. Thus, electing NOT to use an existing benchmark requires a strong justification. I think requiring a different patient population for a specific problem, e.g., for studying respiratory distress in pediatrics or looking at the efficacy of different treatments for sepsis patients, is a sound justification. However, I'm not fully satisfied with "reduced cohorts" as an explanation in a methods paper not concerned with a specific clinical question. Your manuscript does not anywhere indicate that the proposed approach sensitive to data set size (it should work equally well for ~30K vs. ~50K stays). What is more, many clinicians would tell you that combining pediatric and adult populations is undesirable, and a common critique of ML research that uses large MIMIC cohorts for predicting mortality is that they mix multiple causes.
> > >
> > > > Continuous predictions after observing all the data...
> > >
> > > Why is a fixed length look-back window required? Does this imply that your approach can only use a limited history (which would reduce the benefit of using an RNN)?

---

> > > > ### Author Response · Authors · 2018-12-02
> > > > **Updated the paper with your suggestions.**
> > > >
> > > > Thanks again for the helpful comments. I have updated the paper with your suggestions.
> > > >
> > > > Q: Can you further explain "global" here?...
> > > > A: Yes, global is across all data points for that feature in the dataset. For the time series missing entirely, our interpolation model assigns the starting point (time t=0) value of the time series to the global mean before applying the two-layer interpolation network. We will mention this clearly in our final paper.
> > > >
> > > > Q: Experiment with reduced number of cohorts...
> > > > A: We experimented with the benchmarked dataset mentioned in Harutyunyan, et al. [1]. We report the results on the test set below:
> > > >
> > > > Model                 AUC    AUPRC
> > > > LSTM                  0.854   0.516
> > > > IpN + LSTM .     0.861   0.537
> > > > *IpN: Proposed Interpolation Network
> > > > This dataset contains 12 continuous value time series and 47 discrete value time series. Although our interpolation network is most suitable for the continuous value time series, it still achieves slightly better performance on this dataset.
> > > >
> > > > Q:  Continuous predictions after observing all the data...
> > > > A:  What I meant is if we want to make continuous rolling predictions with the same model (i.e. same input size), then the amount of lookback window should be kept fixed.  For example, if there is a model trained with an input window of 24 hrs with given inducing point 1 per hour, then the input to the prediction network would be (batch_size, 24,  features). With such a model, we can make rolling predictions where the fixed lookback window would be 24 hr.
> > > >
> > > > References:
> > > > [1] Harutyunyan, et al. "Multitask learning and benchmarking with clinical time series data." arXiv preprint arXiv:1703.07771 (2017)

---

### Meta-Review · Area_Chair1 · 2018-12-14
**Meta-Review for Interpolation-Predictions paper**

**Confidence:** 4
**Recommendation:** Accept (Poster)

**Metareview:**

After much discussion, all reviewers agree that this paper should be accepted. Congratulations!!